# GAMBAS: Generalised-Hilbert Mamba for Super-resolution of Paediatric Ultra-Low-Field MRI

**Levente Baljer**[*1,2]                               LEVENTE.1.BALJER@KCL.AC.UK
**Ula Briski**[1]                                              ULA.BRISKI@KCL.AC.UK
**Robert Leech**[1]                                        ROBERT.LEECH@KCL.AC.UK
**Niall J Bourke**[1]                                        NIALL.BOURKE@KCL.AC.UK
**Kirsten A Donald**[3]                                 KIRSTEN.DONALD@UCT.AC.ZA
**Layla E Bradford**[3]                                 LAYLA.BRADFORD@UCT.AC.ZA
**Simone R Williams**[3]                             SIMONE.WILLIAMS@UCT.AC.ZA
**Sadia Parkar**[4]                                          SADIA.PARKAR@AKU.EDU
**Sidra Kaleem**[4]                                         SIDRA.KALEEM@AKU.EDU
**Salman Osmani**[4]                                     SALMAN.OSMANI@AKU.EDU
**Sean CL Deoni**[5]                           SEAN.DEONI@GATESFOUNDATION.ORG
**Steven CR Williams**[1]                           STEVE.WILLIAMS@KCL.AC.UK
**Rosalyn J Moran**[†1]                               ROSALYN.MORAN@KCL.AC.UK
**Emma C Robinson**[†2]                             EMMA.ROBINSON@KCL.AC.UK
**František Váša**[†1]                                    FRANTISEK.VASA@KCL.AC.UK

[1] *Department of Neuroimaging, King's College London*

[2] *Department of Biomedical Computing, King's College London*

[3] *Department of Paediatrics and Child Health, University of Cape Town*

[4] *Paediatrics and Child Health, Aga Khan University Hospital, Karachi*

[5] *Bill & Melinda Gates Foundation, Seattle, Washington, USA*

**Editors:** Accepted for publication at MIDL 2025

## Abstract

Magnetic resonance imaging (MRI) is critical for neurodevelopmental research, however access to high-field (HF) systems in low- and middle-income countries is severely hindered by their cost. Ultra-low-field (ULF) systems mitigate such issues of access inequality, however their diminished signal-to-noise ratio limits their applicability for research and clinical use. Deep-learning approaches can enhance the quality of scans acquired at lower field strengths at no additional cost. For example, Convolutional neural networks (CNNs) fused with transformer modules have demonstrated a remarkable ability to capture both local information and long-range context. Unfortunately, the quadratic complexity of transformers leads to an undesirable trade-off between long-range sensitivity and local precision. We propose a hybrid CNN and state-space model (SSM) architecture featuring a novel 3D to 1D serialisation (GAMBAS), which learns long-range context without sacrificing spatial precision. We exhibit improved performance compared to other state-of-the-art medical image-to-image translation models. Our code is made publicly available at https://github.com/levente-1/GAMBAS.

**Keywords:** Mamba, Hilbert curve, MRI, Deep Learning, Image Quality Transfer.

---

[*] Corresponding author

[†] Joint senior authors

## 1. Introduction

High-field (HF) MRI ($>1$ T) is vital for radiation-free neurodevelopmental research and clinical assessment. Unfortunately, the high cost of scanners ($\sim$\$1,000,000 per Tesla; Arnold et al. 2023) and corresponding infrastructural requirements result in significant access inequalities across the globe. When comparing access to imaging via MRI units per million population, over a hundred-fold difference is exhibited between high-income countries (HICs) and low- and middle-income countries (LMICs) (e.g. 51.67 units/million in the US vs 0.48 units/million in Ghana; Jalloul et al. 2023).

Ultra-low-field (ULF) imaging systems ($<0.1$T) present a potential solution to issues of access inequality. By relying on large, permanent magnets, they are significantly cheaper and easier to purchase and run than HF systems (Campbell-Washburn et al., 2019). Additionally, their reduced acoustic noise and open design increase patient compliance, especially in paediatric populations in whom the risk of head motion is naturally higher (Padormo et al., 2023). Unfortunately, these benefits come at the cost of reduced signal-to-noise ratio (SNR) and resolution, which renders images suboptimal for visual reading by radiologists or for many currently available processing methods.

Super-resolution (SR) techniques may help bridge the gap between ULF and HF, with deep learning being particularly well-suited to handle the nonlinear differences in tissue contrast between field strengths. Convolutional neural networks (CNNs) have historically dominated the field of medical image translation (Wang et al., 2022), with adversarial training schemes using an added discriminator further enhancing the perceptual quality of outputs (Armanious et al., 2020). CNNs exploit correlations among neighbouring voxels by extracting local features with compact filters. This reduces model complexity, however it limits the ability to model long-range spatial dependencies present in spatially-distributed brain structures, such as white matter or cerebrospinal fluid (Wang and Wu, 2023).

Vision transformers (ViTs) have shown an increased capacity to capture long-range context. By tokenizing an image into a sequence of patches and computing inter-patch similarity via attention operators, transformers learn to capture spatial correlations among distant regions (Dosovitskiy et al., 2021). Unfortunately, transformers exhibit quadratic model complexity with respect to sequence length (i.e. number of patches), impeding the use of patches small enough to maintain high spatial precision. State-space models (SSMs) offer a more efficient alternative for capturing long-range dependencies. They replace computationally expensive self-attention with linear recurrence, permitting images to be fed into the model as a sequence of individual voxels. Mamba, a variant of traditional SSMs with a powerful selection mechanism, has demonstrated impressive performance in a variety of medical imaging tasks, including segmentation, registration and classification (Heidari et al., 2024).

Here we introduce GAMBAS, **G**eneralised-Hilbert M**amba S**uper-resolution, trained on paired paediatric ULF and HF scans (64mT and 3T, respectively). We fuse elements of CNNs with those of Mamba to increase model sensitivity to local and global context, and add adversarial learning to further increase output realism. We additionally present a novel serialisation approach that mitigates the loss of spatial proximity in Mamba layers. Our results indicate that GAMBAS offers superior performance against state-of-the-art (SOTA) transformer- and SSM-based models.

## 2. Related Work

Several works investigate translation between anisotropic 64mT and isotropic >1T brain MRI. SynthSR (Iglesias et al., 2021) employs a U-Net, trained using synthetic low-resolution scans generated from HF segmentations, to super-resolve images of any contrast and resolution, whereas LF-SynthSR (Iglesias et al., 2023) is a dedicated ULF version of the same network. LoHiResGAN (Islam et al., 2023) is an adversarial model with a ResNet-based generator (He et al., 2016), trained on 37 paired 64mT and 3T scans ($T_1$w and $T_2$w). SFNet (Tapp et al., 2024) is another adversarial model, using a SwinUNETR-V2 (He et al., 2023) generator trained on 30 empirical ULF/HF pairs combined with 60 synthetic pairs generated using a two-channel latent diffusion model.

## 3. Methods

### 3.1. Preliminaries

#### 3.1.1. STATE-SPACE MODELS

The state space model (SSM) is a continuous-time latent state model, which maps a 1D input to a 1D output via a linear ordinary differential equation (ODE). Mamba is a specific SSM variant that discretises the ODE using the zero-order hold (ZOH) technique to produce an efficient recurrent formulation, and adds a selective scan algorithm to filter relevant/irrelevant information (see Appendix A for a derivation of Mamba from continuous-time SSMs). Importantly, Mamba demonstrates an ability to model long-range dependencies on par with that of Transformers, while reducing computational complexity (Gu and Dao, 2023).

As Mamba is designed to process 1D data, its application to vision tasks requires image features of shape $(B, C, H, W, D)$ to be flattened and transposed to $(B, L, C)$, where $L = H \times W \times D$. Unlike input sequences in natural language processing tasks, images have no inherent directionality. As such, the approach used to arrange voxels into a 1D sequence determines the way Mamba processes the image and, in turn, affects overall model performance (Zhou et al., 2025).

#### 3.1.2. SPACE-FILLING CURVES

Many approaches to flatten an image rely on a simple 'Raster' scan, which can result in discontinuities when the scan reaches the boundaries of the image. An alternative approach involves the use of space-filling curves, e.g. Z-order curve (Orenstein, 1986) or Hilbert curve (Hilbert 1891; see Figure 1 and Appendix B). Both curves provide a linear mapping between 3D and 1D space that preserves spatial locality (Moon et al., 2001), with the Hilbert curve demonstrating superior order-preserving behaviour (Warren and Salmon, 1993). Because of this property, it has been applied to point-cloud processing in conjunction with transformers (Chen et al., 2022) and, more recently, with Mamba (Zhang et al., 2024). Despite these successes, the fusion of the Hilbert curve with Mamba has not yet been demonstrated for medical image applications. Here, we apply a generalised variant of the Hilbert curve ("Gilbert"; Červený 2019), capable of handling evenly-sided 3D data of any size, to serialise structural MRI input into a 1D vector.

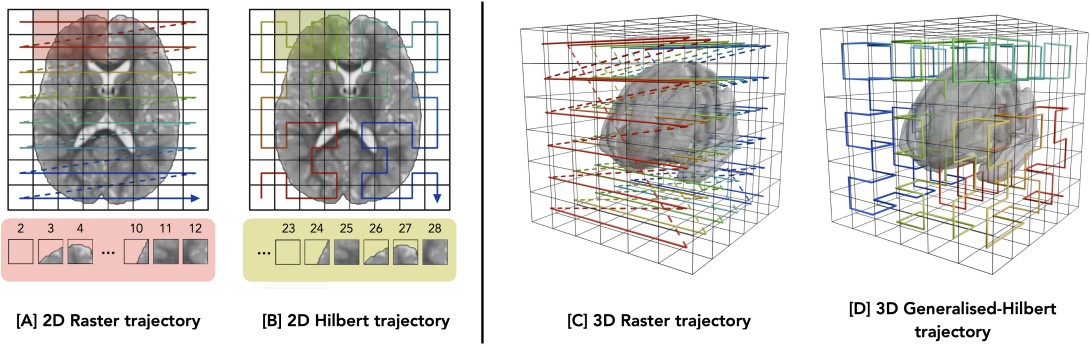

Figure 1: **Left**: [A] 2D Raster trajectory and [B] 2D Hilbert trajectory, with corresponding 2D → 1D serialisation of highlighted region shown below.
**Right**: [C] 3D Raster trajectory and [D] 3D Generalised-Hilbert trajectory.

## 3.2. Architecture

### 3.2.1. GAMBAS

We use a 3D counterpart of the 9-block ResNet architecture, originally introduced in Zhu et al. 2017, as the backbone of GAMBAS. The model follows an encoder-decoder pathway (composed of 3 layers each) with an extended central bottleneck (consisting of 9 layers). The order of operations across layers is kept identical to the original paper, however all 2D convolutions are switched to their 3D counterparts, and batch normalisation is substituted with instance normalisation. Inspired by earlier work incorporating SSMs for 2D multi-modal medical image synthesis (Atli et al., 2024), we insert Mamba blocks into the 9-layer bottleneck, with the positioning determined via ablations studies (see Figure 2 and Appendix D). Through these blocks, encoder outputs are serialised via a generalised Hilbert trajectory, then fed into bidirectional Mamba modules. Outputs from Mamba are then concatenated with earlier CNN outputs and fed into channel-mixing blocks (Dalmaz et al., 2022) to consolidate contextual features from Mamba with structural features from CNNs. Channel-mixing blocks consist of two parallel convolutional branches of varying kernel sizes, mixing features at both identical and differing spatial locations (Tolstikhin et al., 2021).

### 3.2.2. Training objective

We implement GAMBAS as an adversarial model, using PatchGAN as our discriminator (Isola et al., 2017). Once again, we keep operations identical to the original paper, switching only 2D convolutions to 3D convolutions and batch normalisation to instance normalisation. The discriminator has two input channels, taking in both an ULF scan and either a real or synthetic HF scan at each training step to determine whether the input pair is real. As such, our model follows the conditional GAN (cGAN) training objective, summarised by:

$$\mathcal{L}_{cGAN}(G, D) = \mathbb{E}_{x,y}[\log D(x, y)] + \mathbb{E}_x[\log 1 - D(x, (G(x))] \tag{1}$$

where generator $G$ and discriminator $D$ aim to a minimise and maximise expectation $\mathbb{E}$, respectively. To ensure subject anatomy is maintained on conditional image generation, $G$ is tasked with an additional loss of minimising the L1 distance between model outputs and reference HF images:

$$\mathcal{L}_{L1}(G) = \mathbb{E}_{x,y}[\|y - G(x)\|_1] \tag{2}$$

Combining Eq.(1) and Eq.(2) yields the following minimax objective:

$$G^* = \arg\min_G \max_D \lambda_{adv}\mathcal{L}_{cGAN}(G, D) + \lambda_{L1}\mathcal{L}_{L1}(G) \tag{3}$$

Here, $\lambda_{adv} = 1$ and $\lambda_{L1} = 100$ are loss weighting terms selected based on favourable results from earlier research (Isola et al., 2017), (Dalmaz et al., 2022).

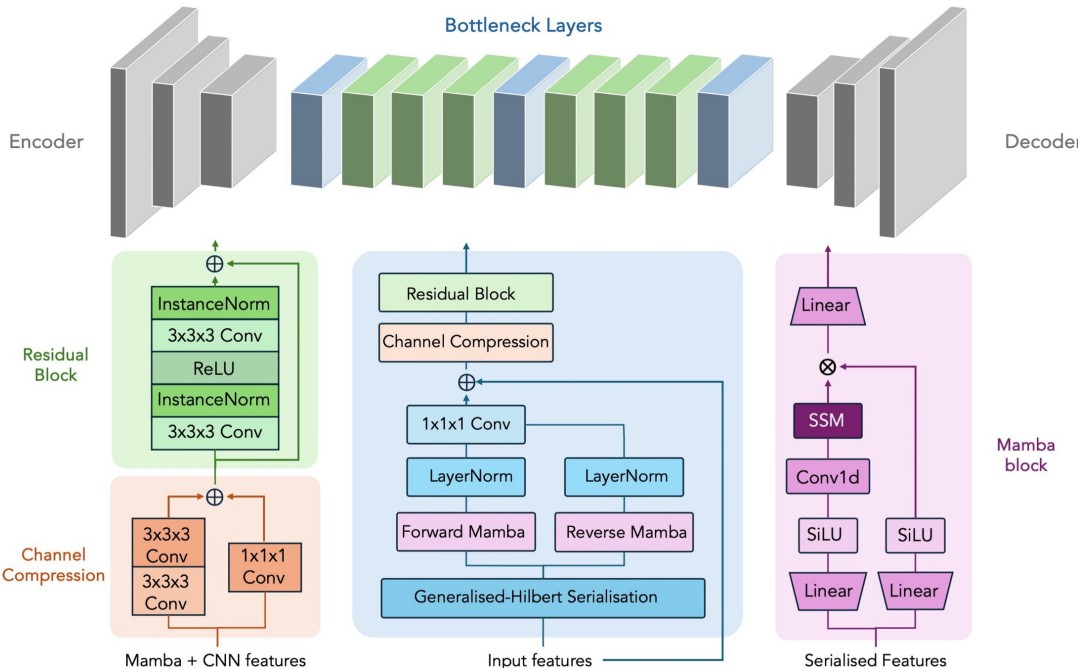

Figure 2: Generator Architecture. Bottleneck layers 1, 5 and 9 (light blue) include forward and reverse Mamba blocks, channel compression and residual blocks. Remaining bottleneck layers (light green) are comprised of a residual block only.

### 3.3. Experiments

#### 3.3.1. DATA AND TRAINING

MRI data used in this paper stem from two separate studies on neurodevelopment over the first 3 years of life in LMICs. The Khula Study (Zieff et al., 2024) investigates the development of executive function in children based in South Africa and Malawi. Children with no known neurological abnormalities were scanned at ages of 3, 6, 12, 18, or 24 months.

Subjects attended two scanning sessions, with HF $T_2$w scans ($1\times1\times1$mm) acquired using a Siemens 3T scanner (Erlangen, DE) and ULF $T_2$w scans ($1.5\times1.5\times5$mm) acquired using a Hyperfine Swoop 64mT system (Guildford, CT). MINE (Maternal and environmental Impact assessment on Neurodevelopment in Early childhood years; Surani et al. 2023) assesses children based in Karachi, Pakistan, beginning at ages of 1, 3, or 6 months and continuing via yearly follow-ups. Subjects also attended two scanning sessions, with HF $T_2$w scans ($0.5\times0.5\times0.5$mm) acquired using a Toshiba 3T scanner (Otawara, JP) and ULF $T_2$w scans ($1.6\times1.6\times5$mm) also acquired using a Hyperfine Swoop 64mT system (Guildford, CT).

Images were assessed for motion or banding artifacts; those with severe distortions were excluded, resulting in a total of 215 paired ULF and HF scans (116 Male, 99 Female; mean age $10.9 \pm 6.7$ months). Pre-processing involved rigid registration (Avants et al., 2009) of all HF scans to a custom age-specific HF template, and of each subject's ULF scan to the corresponding pre-registered HF scan. We additionally resampled all ULF and HF scans to $1\times1\times1$mm resolution to ensure consistency across scanners and sites. The data were split into training/validation/test sets of 137/35/43 scans (i.e. $\sim$64/16/20%), balancing for age and sex, and our model was trained for 1000 epochs on a Nvidia RTX 3090 24GB GPU. We used the Adam optimizer with $\beta_1 = 0.5$, $\beta_2 = 0.999$ and a learning rate of 0.0002. Patch-wise training was conducted, extracting random $128\times128\times128$ patches from input and target images at each training step, and batch size was set to 1. During training, image augmentation was carried out on the fly, including random 3D rotation, BSpline deformation, random flip, and contrast and brightness adjustment.

### 3.3.2. COMPETING METHODS

To compare performance of our model to similar generator architectures, we trained three additional models from scratch using our data: SwinUNETR-V2 (He et al., 2023), ResViT (Dalmaz et al., 2022), and U-Mamba (Ma et al., 2024). These networks all combine local and global context; SwinUNETR-V2 merges the U-Net architecture with Swin-transformers, ResViT adds pre-trained vision transformers into bottleneck layers of a 9-block ResNet, and U-Mamba adds SSM blocks to each encoding layer of the U-Net. As ResViT was not designed to handle 3D data, we provide a 3D implementation here (details in Appendix C). For a fair comparison, all competing methods were implemented as adversarial models with a PatchGAN discriminator, trained via the loss function described in Eq.(3). The quality of model outputs relative to ground-truth HF images is evaluated via normalised root mean squared error (NRMSE), peak signal-to-noise ratio (PSNR), structural similarity index measure (SSIM), and learned perceptual image patch similarity (LPIPS).

We additionally benchmark against other existing ULF SR models; LF-SynthSR and SFNet. LoHiResGAN does not have publicly available weights for a model pre-trained on $T_2$w scans, and thus is excluded from benchmarking. As LF-SynthSR exclusively outputs $T_1$w MRI scans regardless of the contrast of the input, we compare outputs of GAMBAS and SFNet with those of LF-SynthSR via Dice overlap of segmented brain regions, using segmentations obtained via SynthSeg (Billot et al., 2023), a toolkit that is agnostic to contrast and resolution. Segmentation outputs for subjects at 3 months of age and younger (n=10) were highly inconsistent, therefore these were excluded from analyses. Unless otherwise specified, all significance testing was carried out via the Wilcoxon signed-rank test.

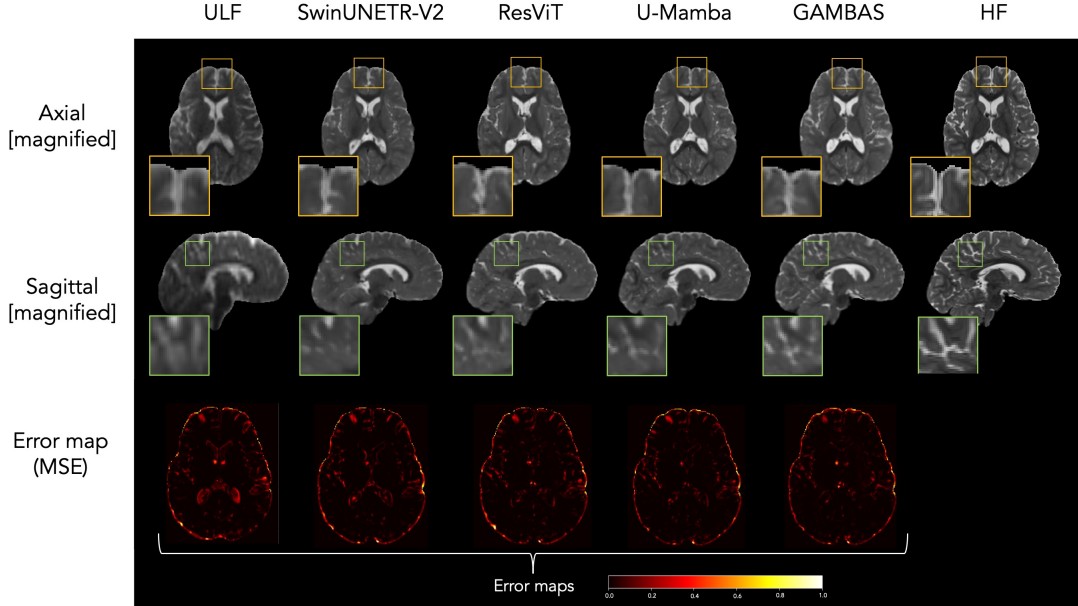

Figure 3: Model outputs and error maps from a single test subject. Left to right: raw ULF scan, SwinUNETR-V2, ResViT, U-Mamba, GAMBAS, reference HF scan.

## 4. Results

From our comparative analyses, we demonstrate that GAMBAS produces visually superior outputs compared to other models assessed (Figure 3). We highlight this with improved performance in all voxelwise metrics compared to SwinUNETR-V2, ResViT, and U-Mamba (Table 1), with GAMBAS improving ULF scores on average by 23.9%. Taking a more overarching view of the results, we additionally observe a pattern whereby Mamba-infused models (U-Mamba and GAMBAS) outperform transformer-infused models (SwinUNETR-V2 and ResViT), potentially signifying that Mamba is better suited for capturing global context in this image translation task. When assessing our performance against existing ULF SR models, segmentation-based analyses reveal that GAMBAS outperforms the only publicly available SR toolkits for use on 64mT ULF scans (LF-SynthSR and SFNet). Across all four tissue types (GMC, GMS, WM, CSF), GAMBAS demonstrates closest correspondence in Dice overlap to HF scans ($\mu$=0.788); higher than both SFNet ($\mu$=0.691) and LF-SynthSR ($\mu$=0.693); see Table 2 and Appendix E.

## 5. Discussion

Our novel GAMBAS network presents an effective and reliable method for translation of ULF to HF MRI, underscoring the potential for deep learning tools in widening accessibility to MRI in resource-limited settings. We demonstrate improved performance compared to other SOTA methods, highlighting the benefits conferred by our approach of fusing novel

Table 1: Image quality metrics for ULF scans and all SR methods trained using our dataset. Asterisk reflects a significant difference compared to all previous methods.

| Model | NRMSE ($\downarrow$) | PSNR ($\uparrow$) | SSIM ($\uparrow$) | LPIPS ($\downarrow$) |
|---|---|---|---|---|
| none [ULF] | 0.441±0.038 | 26.772±1.372 | 0.896±0.021 | 0.0402±.0075 |
| SwinUNETR-V2 | 0.315±0.044 | 29.750±1.827 | 0.911±0.017 | 0.0231±.0047 |
| ResViT | 0.310±0.046 | 29.885±1.811 | 0.911±0.016 | 0.0235±.0046 |
| U-Mamba | 0.294±0.043 | 30.336±1.868 | 0.914±0.016 | 0.0226±.0050 |
| GAMBAS | **0.290±0.044** | **30.469±1.919** | **0.916±0.016*** | **0.0220±.0051*** |

Table 2: Dice coefficients for four tissue types: cortical grey matter (GMC), subcortical grey matter (GMS), white matter (WM) and cerebrospinal fluid (CSF), for input ULF scans, SynthSR and GAMBAS.

| Model | GMC | GMS | WM | CSF |
|---|---|---|---|---|
| none [ULF] | 0.682±0.055 | 0.680±0.216 | 0.698±0.097 | 0.423±0.069 |
| LF-SynthSR | 0.716±0.032 | 0.840±0.102 | 0.778±0.025 | 0.439±0.062 |
| SFNet | 0.731±0.036 | 0.794±0.083 | 0.759±0.042 | 0.478±0.061 |
| GAMBAS | **0.806±0.021*** | **0.899±0.014*** | **0.815±0.017*** | **0.631±0.051*** |

Generalised-Hilbert Mamba blocks with CNNs for image generation. Although U-Mamba also presents a network combining CNN and Mamba modules, they rely on a simple 'Raster' trajectory for 1D serialisation and do not include channel mixing blocks between CNN and Mamba outputs.

By training models with a paediatric dataset, we tackle a particularly challenging image translation task. Our data includes children scanned at various stages of brain development, from early stages of myelination with inverted GM:WM signal intensities relative to adult contrast (Dubois et al., 2021), through a period of strong overlap in tissue intensity (Bui et al., 2020), to later stages of myelination (1 year and above) when adult contrast is established. In spite of this, we demonstrate improved NRMSE, PSNR, SSIM, and LPIPS on a test set spanning subjects across all stages. One notable limitation of our study stems from our use of SynthSeg (Billot et al., 2023) for segmentation. It was chosen as it is the only widely available segmentation toolkit agnostic to contrast and resolution (allowing direct comparison between anisotropic $T_2$w ULF scans, 1mm isotropic $T_2$w GAMBAS/SFNet outputs, and 1mm isotropic $T_1$w SynthSR outputs). However, as SynthSeg was trained on adult data, it performs poorly on scans deviating too much from the normal adult contrast, necessitating the exclusion of 3-month-old subjects from segmentation-based analyses.

In summary, GAMBAS shows great promise for SR of ULF MRI. Future work will explore its application to other datasets, including adult and/or clinical populations.

## Acknowledgments

We would like to acknowledge funding from Artificial Intelligence Methods for Low Field MRI Enhancement, Bill and Melinda Gates Foundation (INV-032788) and Wellcome Leap 1kD programme (The First 1000 Days) [222076/Z/20/Z].

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

## Appendix A. S4 and Mamba

The state space model (SSM) is a continuous-time latent state model, which maps a 1D input $x(t) \in \mathbb{R}^L$ to an output $y(t) \in \mathbb{R}^L$ through hidden state $h(t) \in \mathbb{R}^N$ via the following linear ODE:

$$
\begin{aligned}
h'(t) &= \mathbf{A}h(t) + \mathbf{B}x(t), \\
y(t) &= \mathbf{C}h(t) + \mathbf{D}x(t)
\end{aligned}
\tag{4}
$$

Here, $\mathbf{A} \in \mathbb{R}^{N \times N}$, $\mathbf{B} \in \mathbb{R}^{N \times 1}$, and $\mathbf{C} \in \mathbb{R}^{1 \times N}$ are learnable parameters and $\mathbf{D} \in \mathbb{R}^1$ represents a residual connection. The continuous-time SSM has prohibitive computation and memory requirements for deep learning applications, however, it can be discretized using a timescale parameter $\mathbf{\Delta}$. For this, the zero-order hold (ZOH) technique is employed, whereby continuous parameters $\mathbf{A}$, $\mathbf{B}$ are discretized as $\overline{\mathbf{A}} = \exp(\mathbf{\Delta A})$, $\overline{\mathbf{B}} = (\mathbf{\Delta A})^{-1}(\exp(\mathbf{\Delta A}) - \mathbf{I}) \cdot \mathbf{\Delta B}$. By substituting these into Eq.(4), we obtain the structured state-space sequence model, or S4 (Gu et al., 2022), represented in the following recurrent form:

$$
\begin{aligned}
h_t &= \overline{\mathbf{A}}h_{t-1} + \overline{\mathbf{B}}x_t, \\
y_t &= \mathbf{C}h_t + \mathbf{D}x_t
\end{aligned}
\tag{5}
$$

Given an input $x \in \mathbb{R}^{1 \times L}$, where $L$ is the sequence length, the model can also be computed by convolving a structured convolution kernel $\overline{\mathbf{K}} = (\mathbf{C}\overline{\mathbf{B}}, \mathbf{C}\mathbf{A}\overline{\mathbf{A}\mathbf{B}}, ..., \mathbf{C}\overline{\mathbf{A}}^{\mathbf{L}}\overline{\mathbf{B}})$ across $x$. Although efficient for training, S4 is unable to select data in an input-dependent manner.

Mamba overcomes this hurdle and distinguishes itself from earlier SSMs by setting the model parameters to be input-dependent, endowing the model with a selection mechanism that can filter out irrelevant information and remember relevant information indefinitely (Gu and Dao, 2023). This no longer allows for the convolutional formulation to be applied (as input-dependent parameters prevent the use of a single structured kernel), however efficient training is ensured via a hardware-aware algorithm that computes model outputs using a parallel scan operation.

## Appendix B. Hilbert curve and implementation

The Hilbert curve can be expressed concisely using a Lindenmayer system, or L-system. L-systems consist of three components: 1) an "alphabet" of symbols that form a string, 2) an initial "axiom" string, and 3) a set of "production rules" that describe how symbols are replaced to form a larger string. Crucially, the "alphabet" can be sub-divided into "variables" (symbols that can be replaced) and "constants" (symbols that cannot be replaced). Once a string evolves through a number of successive iterations from the axiom via the production rules, the output is used as a set of instructions for generating a geometric structure. The L-system describing the Hilbert curve can be summarised as:

$$
\begin{aligned}
\textbf{Variables} &= A, B \\
\textbf{Constants} &= F, +, - \\
\textbf{Axiom} &= A \\
\textbf{Production rules} &= A \rightarrow +BF - AFA - FB+, \\
&\quad\; B \rightarrow -AF + BFB + FA-
\end{aligned} \tag{6}
$$

where $F$ is forward, $-$ and $+$ indicate a 90° turn clockwise and counterclockwise, respectively, and $A$ and $B$ are ignored during drawing. The first three iterations of this L-system are depicted in Figure 4.

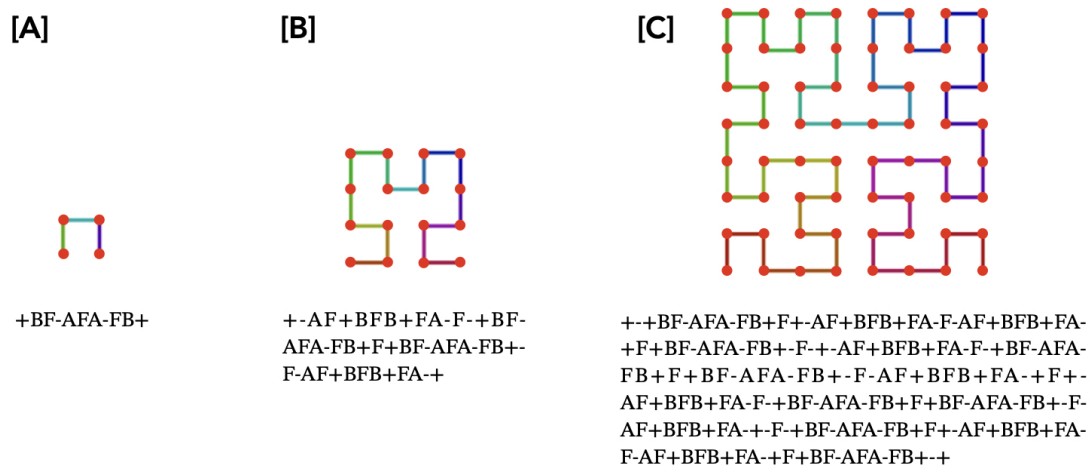

**[A]**

+BF-AFA-FB+

**[B]**

+-AF+BFB+FA-F-+BF-
AFA-FB+F+BF-AFA-FB+-
F-AF+BFB+FA-+

**[C]**

+-+BF-AFA-FB+F+-AF+BFB+FA-F-AF+BFB+FA-
+F+BF-AFA-FB+-F-+-AF+BFB+FA-F-+BF-AFA-
FB+F+BF-AFA-FB+-F-AF+BFB+FA-+F+-
AF+BFB+FA-F-+BF-AFA-FB+F+BF-AFA-FB+-F-
AF+BFB+FA-+-F-+BF-AFA-FB+F+-AF+BFB+FA-
F-AF+BFB+FA-+F+BF-AFA-FB+-+

Figure 4: [A] Iteration 1, [B] iteration 2, and [C] iteration 3 of the Hilbert curve, following starting axiom $A$. The output string is shown on the bottom, and the corresponding geometric structure on top.

The original formulation of the Hilbert curve only produces structures composed of $4^n$ vertices to fill regions in 2D space (see Figure 4). By using a generalised formulation of the Hilbert curve (Červený, 2019), we instead are able to handle 3D image features of any even dimensions. This implementation uses a recursive mechanism to split a cuboid into multiple regions, with the function calling on itself until a trivial path can be computed. Calling this function in each training step would compromise efficiency, however we circumvent this by only using the function once upon model initialization, at which point a tensor is stored with indices that delineate the serialisation of 3D inputs with pre-specified dimensions. We operate with patches of size $128 \times 128 \times 128$, which are then downsampled through our encoder blocks to be of size $32 \times 32 \times 32$. As such, a Generalised Hilbert-curve for a cube with height, width and depth of 32 is computed and stored upon model initialisation, and is used in each training step to serialise 3D image features before being fed into forward and reverse Mamba blocks (Figure 2).

## Appendix C. Implementation of competing methods

### C.1. SwinUNETR-V2

SwinUNETR-V2 was implemented using the MONAI framework (Cardoso et al., 2022), with all parameters (i.e. network feature size, number of layers in each stage and number of attention heads) set to default settings, in line with the original paper (He et al., 2023).

### C.2. U-Mamba

As U-Mamba is implemented in the self-adapting nnU-Net (Isensee et al., 2018) framework, which is tailored for segmentation data, we examined the network configurations for various datasets listed in the original paper (Ma et al., 2024) and selected the one most closely aligned with our own data (3D Abdomen MRI). Consequently, we used the "U-Mamba_Enc" variant (employing Mamba blocks in all encoder layers), with 6 encoder/decoder stages and pooling per axis set at (3, 5, 5).

### C.3. ResViT

ResViT used an identical 9-block ResNet backbone as GAMBAS, however transformer modules were inserted into bottleneck layers 1 and 4, in accordance with the original paper. These modules used weights from transformer component of the ImageNet-pretrained model R50+ViT-B/16 (https://github.com/google-research/vision_transformer). As the pre-trained transformer expects 2D inputs of size 16×16, we downsampled image features in transformer blocks to match the desired height and width, and performed patch embedding across the depth dimension.

## Appendix D. Ablation studies

### D.1. Insertion of Mamba blocks

Table 3: Ablations on the number of mamba blocks inserted into the bottleneck layers of GAMBAS. (1,5,9) denotes mamba blocks in the first, fifth, and ninth layers.

| Model | NRMSE ($\downarrow$) | PSNR ($\uparrow$) | SSIM ($\uparrow$) | $\mu$ Dice ($\uparrow$) |
|---|---|---|---|---|
| none | 0.314±0.053 | 30.185±1.403 | 0.7795±.0141 | 0.7795±.0339 |
| (5) | 0.312±0.054 | 30.254±1.489 | 0.9149±.0143 | 0.7766±.0352 |
| (1,5,9) | **0.311±0.055** | **30.285±1.482** | **0.9162±.0140** | **0.7802±.0363** |
| (1,3,5,7,9) | 0.315±0.057 | 30.116±1.518 | 0.9153±.0141 | 0.7760±.0336 |
| (1,2,3,...9) | 0.312±0.052 | 30.185±1.432 | 0.9160±.0139 | 0.7782±.0349 |

## D.2. Scanning trajectory

Table 4: Ablation on scanning trajectory used by bidirectional mamba blocks in first, middle, and final bottleneck layers.

| Serialisation | NRMSE (↓) | PSNR (↑) | SSIM (↑) | $\mu$ Dice (↑) |
|---|---|---|---|---|
| Raster | 0.315±0.061 | 30.195±1.598 | 0.9157±.0141 | 0.7782±.0350 |
| Generalised-Hilbert | **0.311±0.055** | **30.285±1.482** | **0.9162±.0140** | **0.7802±.0363** |

## D.3. Unidirectional vs bidirectional mamba

Table 5: Ablation on the use of bidirectional mamba for mamba blocks in first, middle, and final bottleneck layers.

| Model | NRMSE (↓) | PSNR (↑) | SSIM (↑) | $\mu$ Dice (↑) |
|---|---|---|---|---|
| Unidirectional | **0.311±0.052** | 30.265±1.458 | 0.9160±.0140 | 0.7792±.0353 |
| Bidirectional | **0.311±0.055** | **30.285±1.827** | **0.9162±.0140** | **0.7802±.0363** |

## Appendix E. Segmentations

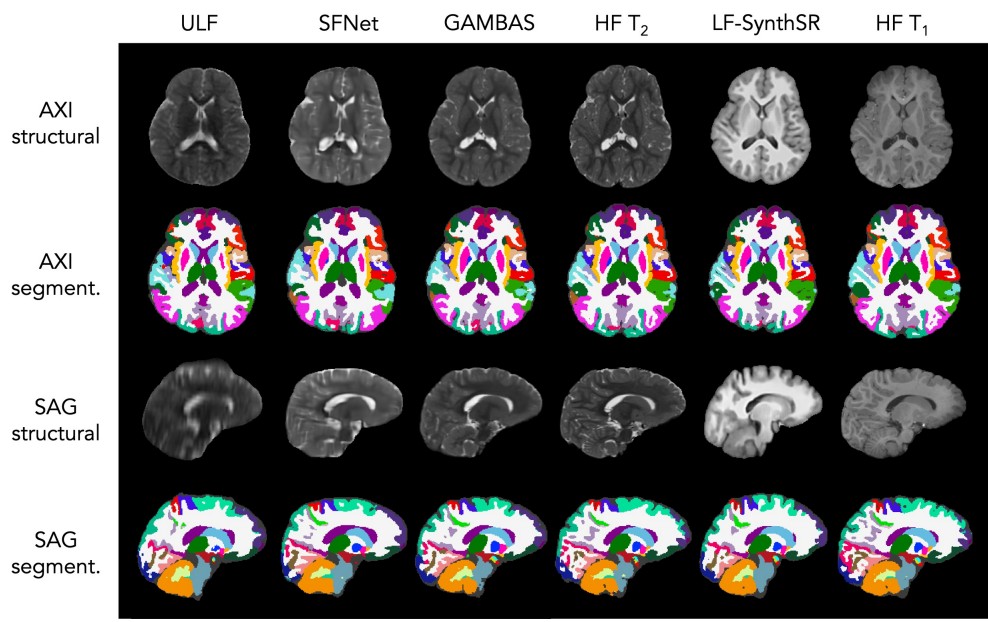

Figure 5: Model outputs and segmentations from a single test subject. Left to right: raw ULF scan, SFNet, GAMBAS, reference $T_2$w HF scan, SynthSR, reference $T_1$w HF scan.

## Appendix F. Bias across sites

Table 6: Model performance across the two imaging sites: South Africa (Khula study) and Pakistan (MINE study). Significance testing, carried out via the Mann-Whitney U-Test, showed no significant differences.

| Imaging site | NRMSE ($\downarrow$) | PSNR ($\uparrow$) | SSIM ($\uparrow$) | $\mu$ Dice($\uparrow$) |
|---|---|---|---|---|
| Khula | 0.298±0.051 | 30.146±2.462 | 0.913±0.016 | **0.789±0.024** |
| MINE | **0.282±0.031** | **30.884±1.272** | **0.917±0.015** | 0.786±0.018 |
| p-value | 0.312 | 0.473 | 0.551 | 0.503 |

