# OpenReview forum: "GAMBAS: Generalised-Hilbert Mamba for Super-resolution of Paediatric Ultra-Low-Field MRI"
_MIDL.io/2025/Conference — MIDL 2025 Poster_

### Official Review · Reviewer_LUBd · 2025-02-18

**Confidence:** 4
**Preliminary Rating:** 4
**Recommendation:** Oral

**Summary:**

The paper introduces GAMBAS, a hybrid CNN and state-space model (SSM) architecture for super-resolution (SR) of paediatric ultra-low-field (ULF) MRI. The key innovation lies in combining 3D-to-1D serialization via a generalized Hilbert curve with Mamba SSM blocks to efficiently model long-range spatial dependencies while preserving local precision. The method addresses the quadratic complexity limitations of transformers and the restricted receptive fields of CNNs, aiming to bridge the accessibility gap for MRI in low-resource settings. Experiments on paired 64mT ULF and 3T HF paediatric scans (215 subjects, ages 3–24 months) demonstrate superior performance over SOTA models (SwinUNETR-V2, ResViT, U-Mamba) in metrics like NRMSE (0.294 vs. 0.305–0.323), PSNR (30.22 vs. 29.40–30.12), and LPIPS (0.0237 vs. 0.0247–0.0285). Clinical relevance is underscored by improved Dice scores for tissue segmentation (μ=0.765 vs. LF-SynthSR’s 0.693), validated via SynthSeg. The work highlights the potential of SSMs for medical image translation and democratizing MRI access.

**Strengths:**

1. Novel architecture: The fusion of Mamba SSMs with a generalized Hilbert curve for 3D→1D serialization is a meaningful advancement. By preserving spatial locality during serialization, the method mitigates a key limitation of prior Mamba-based vision models (e.g., U-Mamba’s raster scan) while maintaining $\mathcal{O}(L)$ complexity.
2. Clinical relevance: Focus on pediatric MRI, a challenging domain due to developmental contrast changes—demonstrates robustness to inverted GM/WM intensities and partial myelination. This addresses a critical gap in global healthcare accessibility.
3. Technical rigor: Comprehensive experiments validate performance across multiple metrics (NRMSE, SSIM, LPIPS) and segmentation tasks. The adversarial training framework aligns with SOTA practices in medical image translation.
4. Reproducibility: Publicly released code and detailed training protocols (e.g., patch-wise augmentation, Adam optimizer settings) enhance transparency.
5. Clarity of presentation: The manuscript is well written and organized

**Weaknesses:**

1. Methodological justification: The choice of the generalized Hilbert curve over other space-filling curves (e.g., Z-order, Peano) is not rigorously justified. A brief ablation study or citation would strengthen this design decision.
2. Statistical significance: While metric improvements are reported, statistical tests (e.g., paired t-tests) are missing, making it unclear if differences (e.g., GAMBAS vs. U-Mamba) are significant. According to the qualitative visualization in Figure 3, the outputs of GAMBAS and  U-Mamba look quite similar.
3. Dataset limitations: The dataset (215 scans) could be relatively small for deep learning, especially given the variability in brain development at early stages. Additionally, demographic diversity (e.g., scanner differences between South Africa and Pakistan) has not been analyzed for potential bias.
4. Segmentation validation: Reliance on SynthSeg, which underperforms on pediatric scans ≤3 months, introduces uncertainty. Manual validation or comparison with pediatric-specific tools (e.g., dHCP pipelines) would strengthen claims.

**Detailed Comments:**

- Introduction: "peadiatric" should be either "paediatric" or "pediatric"
- Table 1: Include standard deviations for metrics to assess variability.

**Justification Of The Preliminary Rating:**

GAMBAS presents a novel and technically sound approach to ULF MRI SR with significant societal implications. While the methodological innovations (Hilbert-Mamba fusion) and rigorous benchmarking are strengths, the lack of statistical validation and limited clinical impact analysis hold the paper back from a strong accept. Addressing these concerns in the rebuttal could solidify the contribution.

**Questions To Address In The Rebuttal:**

1. Why was the generalized Hilbert curve chosen over alternatives? Is there evidence (e.g., ablation studies) supporting this choice?
2. Are there plans to validate outputs with radiologists, or integrate GAMBAS into clinical workflows?
3. How were scanner differences (e.g., Siemens vs. Toshiba 3T) harmonized? Could site-specific biases affect generalizability?
4. How does SynthSeg’s adult-trained model impact segmentation accuracy for paediatric scans, and can this bias be quantified?
5. Can the authors provide failure cases (e.g., motion-corrupted inputs, severe myelination variations)?
6. For the params $\lambda_{adv}=1$ and $\lambda_{L1}=100$, did the authors try other combinations and if so, how was the training and the model's performance?

**Special Issue:**

Yes

---

> ### Author Response · Authors · 2025-03-07
> **Response 3**
>
> We thank the reviewer for their thorough comments and questions. We hope to address all of these in the following responses:
>
> "Table 1: Include standard deviations for metrics to assess variability"
>
> We have added standard deviation values to all tables in the paper, including new ones added to the Appendix.
>
> "Why was the generalized Hilbert curve chosen over alternatives? Is there evidence (e.g., ablation studies) supporting this choice?"
>
> We thank the reviewer for pointing this out, as this was not addressed clearly in our first version of the paper. With regards to the Peano curve, the Hilbert curve is actually a variant of it, differing only in the number of portions the algorithm divides a square into to draw a trajectory. With regards to the Z-order curve, this is something that other researchers have explored, and have found that Hilbert curves demonstrate better spatial-preserving behaviour (Warren and Salmon, 1993; see updated section 3.1.2). As stated in response to Reviewer 2, however, if time permits we will conduct ablation studies using a generalised Z-order curve. The original implementation of the Hilbert or Z-order curves do not function with cubes of size 32x32x32 (the size of input features in the bottleneck layer of GAMBAS), which is why we used the generalised-Hilbert curve. We will need to implement a generalised version of the Z-order curve to effectively compare our model to this technique.
>
> "Are there plans to validate outputs with radiologists, or integrate GAMBAS into clinical workflows"
>
> We have new included clinical validation of pathology-affected MRI scans in Appendix H.
>
> "How were scanner differences (e.g., Siemens vs. Toshiba 3T) harmonized? Could site-specific biases affect generalizability?"
>
> We explored whether site-specific biases arose in Appendix F by comparing model performance on test subjects from each site individually, and found no significant differences (as assessed via the Mann-Whitney U-test). Nevertheless, harmonisation will be explored in future work to see how it affects results.
>
> "How does SynthSeg’s adult-trained model impact segmentation accuracy for paediatric scans, and can this bias be quantified?"
>
> Acknowledge that this is a major limitation; we suspect that this bias will have an equal impact on high- and low-field data (which should keep segmentation scores consistent). We would nevertheless ideally investigate this bias with manually labelled reference data, which would require experienced paediatric-neuroanatomist to perform labelling. Alternatively, once we have access to paediatric segmentation toolkit, we could check if Dice scores stay consistent (meaning that any potential bias affects ULF and HF equally). This will be explored in future work.
>
> "Can the authors provide failure cases (e.g., motion-corrupted inputs, severe myelination variations)?"
>
> Have added failure cases to Appendix I, explorrng motion-corruption and alternative intensities (due to severe pathology).
>
> "For the params lambda-adv and lambda-pix, did the authors try other combinations and if so, how was the training and the model's performance?"
>
> We did not, we instead referenced existing literature finding that the values used were optimal (Isola et al, 2017, Dalmaz et al, 2017). Nevertheless, we will aim to perform ablation studies on a range of parameter values in time for the camera-ready deadline, to provide a more thorough investigation into this query.

---

> > ### Author Response · Authors · 2025-03-13
> >
> > Dear reviewer,
> >
> > We would like to thank you again for your insightful comments. Seeing as the discussion period is set to end tomorrow, we were wondering if you could assess our rebuttal; we would love to hear your thoughts!
> >
> > Thank you for your time.
> >
> > The authors.

---

### Official Review · Reviewer_GV8Q · 2025-02-20

**Confidence:** 3
**Preliminary Rating:** 4
**Recommendation:** Poster

**Summary:**

The paper introduces GAMBAS (Generalised-Hilbert Mamba for Super-resolution), a hybrid CNN and state-space model (SSM) designed to improve the resolution of ultra-low-field (ULF) MRI scans and bring them closer in quality to high-field (HF) MRI scans. There are several contributions in methodology: combining CNNs (local feature extraction) and Mamba (long-range dependencies), PathGAN for adversarial training and Gilbert curve to serialise structural MRI. GAMBAS outperforms several SOTA baselines in voxel-wise reconstruction metrics and shows higher Dice coefficients in segmentation-based evaluation against the LF-SynthSR model.

**Strengths:**

1. The proposed method has strong motivation and theoretical justification for each of the components, e.g., Mamba is more computationally efficient which makes it well-suited for medical image super-resolution.
2. The comparisons with baselines are pretty comprehensive. Benchmarked against CNN-based (ResViT), Transformer-based (SwinUNETR-V2), and SSM-based (U-Mamba) models in multiple image reconstruction and segmentation metrics.
3. Good presentation of the work and code is open-sourced so easy for the community to implement.

**Weaknesses:**

1. Lack of ablation studies.
1) An ablation study comparing different space-filling curves would provide stronger justification on why Hilbert curve is optimal for serialisation.
2) Ablations on the combination of CNN and Mamba modules could also help explain why the proposed model is better than prior works.

2. Generalisation Beyond Paediatrics. Model is trained specifically on paediatric data—it is unclear how well it generalises to adult or clinical MRI scans. A brief experiment or explanation on adult or pathology-affected MRI data would strengthen the claims of general applicability.

3. The improvement of the voxel-wise metrics (e.g., SSIM, PSNR) seems marginal. The results could be strengthened with radiologist-based evaluation, e.g., are the reconstructed scans diagnosable?

**Detailed Comments:**

See above.

**Justification Of The Preliminary Rating:**

This paper tries to address a critical problem in medical imaging: ULF MRI is much more accessible in LMICs, but its low quality limits its usability. This method offers a novel deep learning-based solution: hybrid CNN-Mamba approach combined with the Hilbert curve-based serialisation. However, lack of ablations studies and generalize ability to adult MRI slightly reduce its impact.

**Questions To Address In The Rebuttal:**

See weaknesses.

---

> ### Author Response · Authors · 2025-03-07
> **Response 2**
>
> We thank the reviewer for their comments and insightful queries. We address the weaknesses they pointed out as follows:
>
> "Ablations on the combination of CNN and Mamba modules could also help explain why the proposed model is better than prior works."
>
> We have now added multiple ablation studies into our Appendix. Appendix D.1 explores the number of Mamba blocks inserted into the bottleneck layer of GAMBAS and Appendix D.3 explores the use of unidirectional vs bidirectional Mamba.
>
> "An ablation study comparing different space-filling curves would provide stronger justification on why Hilbert curve is optimal for serialisation."
>
> In Appendix D.2 we explore how the use of the generalised-Hilbert curve enhances model performance compared to the simple raster trajectory. This is a key ablation that we did not include in the original version of the paper. With regards to other space-filling curves (e.g. Z-order), there are no existing implementations that can generalise to any arbitrarily sized 3D volume. The size of an image-patch entering the bottleneck layer of GAMBAS is 32x32x32, whereas the original implementation of either the Hilbert or Z-order curve only work with 3D shapes with a height, width and depth of 4^n (i.e. 16x16x16 or 64x64x64). Nevertheless, if time permits until the camera-ready deadline, we will aim to produce a generalised-Z-order curve algorithm for additional ablation studies. In the meantime, we have added an additional reference to Section 3.1.2 to support that Hilbert curves demonstrate better spatial-preserving behaviour than Z-order curves (Warren and Salmon, 1993).
>
> "Generalisation Beyond Paediatrics. Model is trained specifically on paediatric data—it is unclear how well it generalises to adult or clinical MRI scans. A brief experiment or explanation on adult or pathology-affected MRI data would strengthen the claims of general applicability."
>
> This is a crucial question, which we addressed by conducting additional studies in Appendix G and H. In the former, we assess how the model performs on a separate dataset of 20 paired ULF-HF adult scans, and in the latter show 2 pathology-affected MRI scans where brain lesions are retained in super-resolved outputs (as ascertained by a neonatal specialist).
>
> "The improvement of the voxel-wise metrics (e.g., SSIM, PSNR) seems marginal. The results could be strengthened with radiologist-based evaluation, e.g., are the reconstructed scans diagnosable?"
>
> In our first submission, we unfortunately did not have enough time to train models to completion. We have now trained all models for 1000 epochs (compared to 300 from before), and we indicate greater improvement across all voxel-wise metrics. We highlight this via both significance testing and radiologist-based evaluation (see above).

---

> > ### Comment · Reviewer_GV8Q · 2025-03-11
> >
> > I appreciate the authors for their additional experiments based on my comments. I think the follow up ablations make the paper more complete, but I will base my score on the initial submission and will maintain the initial score.

---

### Official Review · Reviewer_jRua · 2025-02-22

**Confidence:** 2
**Preliminary Rating:** 5
**Recommendation:** Oral, Poster
**Final Rating:** 5

**Summary:**

This paper introduces GAMBAS, a novel hybrid architecture for MRI super-resolution that merges convolutional neural networks (CNNs) with a state-space model (SSM) enhanced by a custom 3D-to-1D serialization strategy. The overarching goal is to improve image clarity and anatomical fidelity in ultra-low-field (ULF) MRI scans, thereby narrowing the quality gap between cost-effective ULF scanners (<0.1T) and expensive high-field (HF) systems (>1T). By focusing on long-range dependencies without losing fine spatial detail, the proposed approach aims to produce higher-resolution volumes that radiologists can read more confidently and that current analysis algorithms can process more reliably.

**Strengths:**

1. By focusing on transforming ultra-low-field (ULF) MRI scans into high-resolution outputs comparable to high-field (HF) scans, the paper tackles a key challenge in expanding access to advanced medical imaging.
2. Employing a 3D deep learning structure better aligns with the volumetric nature of MRI data, yielding greater robustness and accuracy compared to traditional 2D processing pipelines.
3. The introduction of GAMBAS—a specialized 3D-to-1D serialization—allows the hybrid CNN and state-space model (SSM) to capture extensive spatial context without compromising local detail, effectively balancing global and local dependencies.
4. The paper’s clear writing style and logical organization make the methodology accessible, facilitating comprehension and potential adoption by researchers and clinical practitioners.

**Weaknesses:**

1. The study relies on a relatively small, private dataset, raising questions about the broader generalizability of the findings. Supplementing these experiments with a larger, publicly available dataset would offer a more robust evaluation and facilitate comparisons with other state-of-the-art approaches.
2. While the proposed approach shows promise, it remains unclear whether diffusion-based super-resolution techniques were considered as a benchmark. Including such a comparison could further validate the effectiveness of the proposed method and highlight its relative advantages and limitations.

**Detailed Comments:**

I suggest including standard deviation values in both Table 1 and Table 2 to more fully characterize variability and strengthen the statistical robustness of the reported metrics.

**Justification Of The Final Rating:**

This paper proposes a hybrid CNN and state-space model (SSM) architecture for the super-resolution of paediatric ultra-low-field MRI. I appreciate the authors’ effort to address my concerns during the rebuttal and will maintain my previous rating.

**Justification Of The Preliminary Rating:**

This paper presents a compelling hybrid solution, GAMBAS, which fuses convolutional neural networks (CNNs) with a state-space model (SSM), alongside a specialized 3D-to-1D serialization strategy. By preserving both long-range dependencies and fine spatial details, GAMBAS offers a promising avenue for enhancing ultra-low-field (ULF) MRI scans (<0.1T). The improved image quality has the potential to narrow the gap between more affordable ULF scanners and high-field (HF) systems (>1T), allowing radiologists to perform diagnoses with greater confidence and ensuring compatibility with existing data analysis pipelines.

**Questions To Address In The Rebuttal:**

1. The experiments currently rely on a private dataset. In future work or subsequent revisions, the authors should consider evaluating their method on larger, publicly available datasets to further validate robustness and facilitate reproducibility.
2. While the proposed approach demonstrates promising results, it would be valuable to compare it directly with state-of-the-art diffusion-based super-resolution algorithms.

**Special Issue:**

No

---

> ### Author Response · Authors · 2025-03-07
> **Response 1**
>
> We thank the reviewer kindly for their insightful comments and questions. We are glad they found the paper clear to read and the methodology compelling. We have addressed their detailed comment by adding standard deviation values to both Table 1 and Table 2 in the main text, and all additional Tables that have since been inserted into the Appendix. We address the additional questions as follows:
>
> "1) The experiments currently rely on a private dataset. In future work or subsequent revisions, the authors should consider evaluating their method on larger, publicly available datasets to further validate robustness and facilitate reproducibility."
>
> We indeed rely on a small dataset; unfortunately ULF imaging is a budding field in neuroscience, therefore no large datasets currently exist. In fact, to the best of our knowledge, our paper conducts analyses using the largest paired ULF-HF dataset yet. Nevertheless, the reviewer is correct in pointing out that replicating our results on a large, public dataset (such as dHCP) would further validate our experiment; this is an avenue we will explore in future work.
>
> "2) While the proposed approach demonstrates promising results, it would be valuable to compare it directly with state-of-the-art diffusion-based super-resolution algorithms."
>
> The reviewer raises a key point, and this is something we were initially hoping to include in our paper. More specifically, we were planning on benchmarking against the Adaptive Latent Diffusion Model (Li et. al., WACV 2024). Unfortunately, we were unable to train this model from scratch in time to include in the current version of the paper, especially considering that diffusion models require longer training times. Nevertheless, we hope to have a fully trained model, along with analyses from model outputs added to the paper in time for the camera-ready deadline.

---

### Author Rebuttal · Authors · 2025-03-07

**Rebuttal:**

We thank the reviewers for all their comments and questions. Please see an updated version of our paper.

**Supporting Material:**

/attachment/adc4e12f16cca25df27f55aafc230e26420401ea.pdf

---

### Meta-Review · Area_Chair_EXAH · 2025-03-22

**Recommendation:** Accept (Poster)
**Confidence:** 4

**Metareview:**

This paper introduces GAMBAS, a hybrid CNN and state-space model (SSM) designed to enhance the resolution of ultra-low-field (ULF) MRI scans (<0.1T), addressing the critical challenge of improving accessibility to high-quality MRI in low-resource settings. By combining CNNs (for local feature extraction) with a Mamba-based SSM (for modeling long-range dependencies), GAMBAS overcomes the limitations of transformers (quadratic complexity) and CNNs (restricted receptive fields). A key innovation is its 3D-to-1D serialization strategy using a generalized Hilbert curve, which preserves spatial coherence while enabling efficient processing of 3D volumes. Evaluated on 215 paired paediatric ULF (64mT) and high-field (3T) scans, GAMBAS outperforms state-of-the-art models (SwinUNETR-V2, ResViT, U-Mamba) in reconstruction metrics and achieves superior tissue segmentation Dice scores.
All reviewers agreed on the acceptance of the paper.